# Zeaxanthin and Lutein: Photoprotectors, Anti-Inflammatories, and Brain Food

**DOI:** 10.3390/molecules25163607

**Published:** 2020-08-08

**Authors:** Barbara Demmig-Adams, Marina López-Pozo, Jared J. Stewart, William W. Adams

**Affiliations:** Department of Ecology & Evolutionary Biology, University of Colorado, Boulder, CO 80309-0334, USA; marina.lopezpozo@colorado.edu (M.L.-P.); jared.stewart@colorado.edu (J.J.S.); william.adams@colorado.edu (W.W.A.III)

**Keywords:** chronic inflammation, redox regulation, signaling, macular degeneration, *meso*-zeaxanthin, photosynthesis, radiation, thermal dissipation, vision, xanthophyll cycle

## Abstract

This review compares and contrasts the role of carotenoids across the taxa of life—with a focus on the xanthophyll zeaxanthin (and its structural isomer lutein) in plants and humans. Xanthophylls’ multiple protective roles are summarized, with attention to the similarities and differences in the roles of zeaxanthin and lutein in plants versus animals, as well as the role of *meso*-zeaxanthin in humans. Detail is provided on the unique control of zeaxanthin function in photosynthesis, that results in its limited availability in leafy vegetables and the human diet. The question of an optimal dietary antioxidant supply is evaluated in the context of the dual roles of both oxidants and antioxidants, in all vital functions of living organisms, and the profound impact of individual and environmental context.

## 1. Introduction

Carotenoids are found across the tree of life, and include oxygen-free carotenes and oxygen-containing xanthophylls (Figure 1). The focus of this review is a small group of carotenoids essential to humans, that must be obtained from the diet, and include β-carotene (provitamin A), zeaxanthin, and lutein. The synthetic pathway of zeaxanthin formation from β-carotene exists in photosynthetic organisms, ranging from bacteria to plants. Plants, furthermore, synthesize a close structural isomer of zeaxanthin, lutein, from α-carotene. In contrast, humans and other animals lack the synthetic pathway for the de-novo formation of carotenoids but are able to (i) convert carotenoids with provitamin A activity in several steps to retinal that becomes part of the vision process and (ii) convert some lutein to *meso*-zeaxanthin, a stereoisomer of dietary zeaxanthin (Figure 1; [1]). A notable aspect of zeaxanthin metabolism in plants is the additional enzymatic control of its level by conversion among three xanthophylls, in what is known as a xanthophyll cycle; zeaxanthin is rapidly removed by conversion to violaxanthin via the intermediate antheraxanthin (Figure 1); when light levels are low and rapidly reformed from violaxanthin via antheraxanthin; when light levels are high (for more detail, see Section on Zeaxanthin and Lutein in Plants below). Its rapid conversion to violaxanthin makes zeaxanthin scarce in leafy vegetables.

In plants and humans alike, both zeaxanthin and lutein have a number of functions, including photoprotection against damage by intense light, detoxification of oxidants (reactive oxygen species, ROS, and other radicals), and the maintenance of structural and functional integrity of biological membranes. Despite similar qualitative roles, effects differ in degree between zeaxanthin and lutein. Zeaxanthin is the more effective antioxidant, presumably due to its longer system of conjugated double bonds [2,3,4] and has a more pronounced impact on membrane integrity (see below).

Carotenoids can be bound to proteins, and can also be found in the lipid bilayer of biological membranes (Figure 2). In leaves, β-carotene, lutein, and a major fraction of zeaxanthin are bound to specific proteins in the light-collecting complexes of chloroplasts. In addition to its function in the protection of chlorophyll, zeaxanthin also functions in the lipid bilayer of the photosynthetic membrane (or possibly at the protein-membrane interface; [5]). In human tissues, β-carotene, lutein, zeaxanthin, and *meso*-zeaxanthin can be found in the lipid bilayer of membranes, and can also be bound to various types of proteins (e.g., a glutathione *S*-transferase in the human retina binds zeaxanthin and *meso*-zeaxanthin; Figure 2; [6,7]).

Zeaxanthin and lutein both assume a position in biological membranes that is different from β-carotene’s position (Figure 2). Both xanthophylls have polar and nonpolar regions that align with the respective polar and nonpolar regions of phospholipid bilayers, whereas β-carotene is fully nonpolar and dissolves in the membrane core (Figure 2). Zeaxanthin and lutein orient themselves across the membrane, as shown in Figure 2 [8]. Whereas all zeaxanthin and lutein were located within membranes in a single-bilayer system [8], some lutein, but no zeaxanthin, was also found in the inter-membrane region of multiple-bilayer systems [9]. It has been suggested that the intra-membrane position allows xanthophylls to serve as “molecular rivets” that enhance membrane structural integrity [10], as well as decrease the susceptibility of membrane lipids to oxidation [11]. Unlike the xanthophylls with their hydrophilic end groups and hydrophobic middle portion, the fully hydrophobic β-carotene interacts exclusively with the hydrophobic membrane core, and does not interact with the phospholipids’ hydrophilic head groups (Figure 2).

While this review focuses on plants (as an example of an organism capable of de-novo carotenoid synthesis) and humans (as an example of an organism dependent on dietary carotenoid supply), many other organisms also contain zeaxanthin and lutein, as can be illustrated by the following selected examples. Zeaxanthin is present in several groups of bacteria, including one species named “zeaxanthin-making” (*Paracoccus zeaxanthinifaciens*) that was collected from a radioactive site in Japan; these bacteria possessed exceptionally high levels of zeaxanthin (as well as some zeaxanthin-related xanthophylls), and exhibited an unusually high resistance to ionizing radiation, UV light, and hydrogen peroxide [12]. Cyanobacteria, including those partnered with fungi and other bacteria as symbionts in a lichen, accumulate higher levels of zeaxanthin in response to more sunlight-exposed habitats [13,14]. Among plants, zeaxanthin and lutein are responsible for the yellow color of corn kernels, of the marigold (*Tagetes*) flower, and of many species’ yellow leaves during senescence in the autumn [15]. The silkworm *Bombyx mori* is an example of an invertebrate owing its yellow color to the presence of lutein [16]. An example for a non-primate vertebrate with zeaxanthin and lutein can be found in birds. During embryonic development in the egg, zeaxanthin and lutein are moved from the yolk to the developing retina, and the yellow down feathers characteristic of some newly-hatched chickens (for this and additional examples, see [17]).

## 2. The Duality of Light

On the one hand, light/electromagnetic radiation is essential for living organisms – as a direct source of energy and information for photosynthetic producers of chemical energy, and as an indirect source of food and a source of information for many non-photosynthetic organisms. On the other hand, light is potentially threatening, since it gives rise to the production of highly reactive forms of oxygen when interacting with a pigment able to absorb light and pass on excitation energy to ubiquitous oxygen. Just as a painting will bleach during extended exposure to sunlight, leaves of plants as well as eyes and skin can be damaged by intense light. However, unlike the painting, living organisms employ a variety of potent photoprotective mechanisms, involving specialized molecules able to block out light, harmlessly remove excitation energy, detoxify ROS, and/or restore oxidized biomolecules. Remarkably, the molecules that perform photoprotection, and particularly the xanthophylls, carry out several of these specific functions.

## 3. Zeaxanthin and Lutein in Plants

Photosynthetic organisms are the producers of not only macronutrients (carbohydrates, proteins, and fats), but also of many chemicals (phytochemicals) that fulfill essential functions in both photosynthetic and non-photosynthetic organisms. The latter organisms thus depend on a dietary supply of not only calories, but also vitamins and other essential micronutrients like carotenoids with functions, that often involve safe operation in diverse environments. The availability of dietary carotenoids for humans is, in turn, impacted by these compounds’ function in specific environments. For example, leaves produce zeaxanthin only under narrowly defined, specific environmental conditions, and exhibit rapidly fluctuating changes in zeaxanthin content, while they contain stable levels of lutein and β-carotene (see below).

### The Xanthophyll Cycle and Photoprotective Energy Dissipation as Harmless Heat

Exposure of leaves to rising light intensity over the course of the day in a sun-exposed location increases the rate of photosynthesis, up to its saturation point that is reached before peak midday exposure (Figure 3A). Further increases in light intensity beyond this saturation point expose the leaf to more light than it can convert to chemical energy in photosynthesis. At peak light exposure, the percent of absorbed light that is utilized in photosynthesis is thus much lower than early in the morning. Such excess excitation energy could, in principle, lead to the production of harmful ROS. However, all plants possess a potent photoprotective process that harmlessly removes excess excitation energy as thermal energy (heat released into the environment). The sum of photosynthetic utilization and photoprotective thermal dissipation of excess light consumes almost all of the absorbed light (Figure 3A), leaving just a small amount of remaining excitation energy to form ROS, that could inform the plant of its sun-exposed location. The thermal dissipation of excess excitation is catalyzed by zeaxanthin (Figure 3B); leaf zeaxanthin content and thermal dissipation of excess excitation energy increase concomitantly with increasing light intensity. Zeaxanthin is formed from its precursor violaxanthin, via the intermediate antheraxanthin up to peak light intensity, and reconverted to violaxanthin in the afternoon as light levels decline (Figure 3B).

Tight control over the level of zeaxanthin and its engagement in dissipation of excess excitation energy over the course of the day is necessary, because zeaxanthin does not just dispose of unutilized light, but actively competes with photosynthesis for excitation energy, which is why rapidly growing plants remove zeaxanthin when light availability is limiting to photosynthesis. This basic fact has consequences for the ability of consumers of plants to acquire dietary zeaxanthin. In contrast, lutein and β-carotene are constitutively present (Figure 3B), and thus easier to obtain from leafy greens.

There are some natural environments in which zeaxanthin levels and thermal dissipation of excess do not go strictly parallel to each other. One of these is a shaded environment under a tree canopy with intermittent brief increases in light intensity (sunflecks) as the sun moves over the canopy [20]. Leaves exposed to such a fluctuating light environment form zeaxanthin in the morning, and alternate rapidly between engagement and disengagement of existing zeaxanthin in thermal dissipation of excitation energy [20], via physical changes in the light-harvesting system [21]. We provided proof-of-concept that this behavior of plants can be used to retain zeaxanthin in leaves during periods of low light, by employing fluctuating light regimes under controlled conditions [22].

The mechanistic roles of lutein, zeaxanthin, and tocopherol (vitamin E) interact to allow the control of various excited states formed in photosynthesis. The energy of singlet-excited chlorophyll can either be (1) used in photosynthesis, (2) converted to harmless heat energy by zeaxanthin, or (3) enter into an alternate excited state. Singlet-chlorophyll de-excitation by zeaxanthin can occur by either of two photophysical mechanisms, i.e., pure excitation transfer from singlet chlorophyll to zeaxanthin, or de-excitation by rapid, reversible exchange of an electron (charge transfer) between the two molecules [23,24]. Any remaining singlet-excited chlorophyll converts to another excited chlorophyll state, triplet-excited chlorophyll, able to transfer its excitation energy to oxygen, which has the potential to interact with molecular oxygen resulting in the formation of the ROS singlet oxygen. Lutein is active in converting the energy of triplet chlorophyll to harmless heat [25], and zeaxanthin can further contribute to this mechanism as well [26]. Singlet oxygen formed in photosynthesis can be de-excited, and radicals based on ROS or oxidized membrane lipids can be reduced, by zeaxanthin and vitamin E ([5]; see also Section below on Synergy between Xanthophylls and Other Antioxidation Systems).

## 4. Zeaxanthin and Lutein in Humans

As stated above, humans and other animals cannot synthesize carotenoids de novo, and must obtain carotenoids (or their precursors) from the diet (see [17]). In humans, the best-studied role of carotenoids is in vision. Notably, β-carotene has vitamin A activity, and can serve as a precursor (Figure 1) to the cleavage product retinal, an essential component of vision purple (rhodopsin). Much of the research on zeaxanthin and lutein in humans has thus far focused on their role in photoprotection against damage, especially age-related blindness (macular degeneration; for reviews, see [7,17]). We present below an overview of the photoprotective role of zeaxanthin and lutein in humans, followed by a segment on evidence for a role of zeaxanthin and lutein in visual and auditory processing, general mental acuity, and in a number of chronic diseases and disorders beyond eye disease.

### 4.1. Transport and Storage of Carotenoids

Carotenoids are transported from the gut to various organs through the blood stream via lipoproteins. Lutein and zeaxanthin are associated mainly with high-density lipoprotein (HDL), which is thought to be the main vehicle for delivery of these xanthophylls to the retina [27]. Lutein and zeaxanthin are also found in human skin [28], serving in protection against UV damage. Skin swelling and cell-division acceleration in response to UV exposure were ameliorated by dietary lutein and zeaxanthin [29]. Furthermore, the liver is the site of β-carotene conversion to vitamin A, as well as a depot for storage of carotenoids [30]. Unborn children receive zeaxanthin and lutein from their mothers throughout the prenatal period up to birth, with xanthophyll levels in the retina and brain of human infants impacted by maternal carotenoid status [17].

### 4.2. Protection of the Eye against Intense Visible Light

The photoprotective role of xanthophylls in the eye involves several mechanisms, including attenuation of blue light in a sunscreen-like function, as well as the removal of ROS by de-excitation [31], as well as the reduction of other reactive radicals [17]. The role of zeaxanthin and lutein in photoprotection of the eye is supported by correlative evidence, as well as experimental or trial-based manipulation. (i) The central portion (macula) of the retina that receives the most intense light features the highest concentration of zeaxanthin and lutein. (ii) Experimental induction of eye damage (photoreceptor death) in a bird model, the Japanese quail *Coturnix japonica*, established that photoreceptor death by exposure to intense light, can be prevented by dietary zeaxanthin [32] or dietary zeaxanthin and lutein [30,33]. (iii) In human clinical trials, individuals receiving zeaxanthin/lutein supplements experienced less vision loss over time than controls (for a review, see Bernstein et al. [17])

There is, furthermore, evidence for a greater photoprotective capacity of zeaxanthin compared to lutein (for reviews, see [7,17,34,35]). The ratio of zeaxanthin (and *meso*-zeaxanthin) to lutein is highest in the macula where the strongest light is received, and lowest in the peripheral, low-light-vision regions of the eye [36,37]; the ratio of zeaxanthin to lutein increases from the diet to the retina; and a portion of dietary lutein is converted to *meso*-zeaxanthin, a stereoisomer of zeaxanthin (Figure 1). This preference for zeaxanthin has been suggested to be due to a greater antioxidant capacity and membrane-stabilizing function of zeaxanthin (and *meso*-zeaxanthin) compared to lutein [2,3]. More research is needed to explore whether the employment of lutein alongside zeaxanthin and *meso*-zeaxanthin is related to the dramatically limited dietary availability of zeaxanthin compared to lutein (see Section above on Zeaxanthin and Lutein in Plants), or whether lutein has any roles that zeaxanthin and *meso*-zeaxanthin cannot fulfill. Attenuation of blue light is presumably accomplished by all three isomers, since xanthophylls absorb strongly between 350 and 500 nm [38].

### 4.3. Enhancement of Eye and Brain Function beyond Disease

It is noteworthy that the macular xanthophylls also enhance the function of the healthy eye—via improved contrast sensitivity [39,40] and reduced glare, which may be related to the filtering of blue light (see [17]). In addition, zeaxanthin and lutein enhance the processing of visual signals in the brain [41] as well as the processing of auditory signals [42]. Recent evidence, furthermore, indicates that zeaxanthin and lutein function broadly in a number of brain regions that are associated with visual perception, cognition, decision-making, and motor coordination [43]. A correlative study [44] reported lower Alzheimer’s mortality in individuals, with higher levels of zeaxanthin, lutein, and lycopene, but not of other carotenoids (β-carotene, α-carotene, or the intermediate β-cryptoxanthin in the synthetic pathway from β-carotene to zeaxanthin; see Figure 1). A recent nutritional intervention with Alzheimer’s patients, consisting of supplementation with either (1) zeaxanthin, *meso*-zeaxanthin, and lutein, or with (2) the three latter xanthophylls plus two omega-3 fatty acids (docosahexaenoic acid and eicosapentaenoic acid) resulted in a greater increase in blood xanthophyll concentrations, and greater caregiver-reported improvements in memory, sight, and mood of patients [45]. More research is needed into the possible effects of xanthophylls and their interaction with omega-3 fatty acids, with respect to the solubility and bioavailability of the largely hydrophobic xanthophylls and their function in the brain (see also Section below on Synergy between Xanthophylls and Other Antioxidation Systems).

### 4.4. Xanthophylls as Anti-Inflammatory Agents

Age-related macular degeneration and Alzheimer’s disease are both pro-inflammatory diseases involving immune system dysfunction and uncontrolled inflammation. The realization that a state of chronic low-grade inflammation plays a key role in a host of additional chronic diseases (e.g., cardiovascular disease, diabetes, certain cancers, autoimmune diseases) and disorders (e.g., anxiety, depression, bipolar disease, schizophrenia, post-traumatic stress disorder) has been called “one of the most important scientific discoveries in health research in recent years” [46]. Strikingly, memory, attention, learning, and overall cognitive performance is also impaired by systemic inflammation—even in adults considered healthy and with no diagnosis of a disease or disorder ([47]; for more detail, see below). One can hypothesize that carotenoids with antioxidant and anti-inflammatory functions may ameliorate some or all of these conditions. There is a substantial body of evidence for correlations between higher carotenoid levels and a lower risk for various pro-inflammatory diseases [48].

Like intense visible light, ionizing radiation (that increases in the upper atmosphere and especially in outer space) produces multiple ROS that can trigger system-wide uncontrolled inflammation [49,50]. There is correlative evidence for a possible role of zeaxanthin in protection against ionizing radiation; pilots who reported consuming more antioxidants, including zeaxanthin and lutein, exhibited less inflammation [51]. Sufficient dietary zeaxanthin and lutein may be of particular concern for future human spaceflight and the associated radiation exposure, and a reliable food source that can be grown in limited space with minimal resources and delivers high levels of zeaxanthin and lutein may be critical [52].

While more research is needed to assess the question of causality between xanthophylls and a lessening of systemic inflammation, evidence for a causal relationship is beginning to emerge from manipulative studies in animal models and humans. For example, Zhou et al. [53] reported that long-term zeaxanthin supplementation lowered the levels of pro-inflammatory hormones and lessened diabetic symptoms, anxiety, and depression in diabetic rats. Stringham et al. [54] extended these findings to humans and, in particular, healthy young subjects that received a supplement with zeaxanthin, *meso*-zeaxanthin, and lutein for six months. Outcomes of this trial included significantly lower levels of pro-inflammatory hormones and enhanced cognitive performance on a variety of complex tasks, including processing speed, as well as various aspects of memory and attention. Similar improvements of cognitive function in young healthy adults as a result of supplementation with zeaxanthin and lutein were reported by Renzi-Hammond et al. [55].

## 5. Synergy between Xanthophylls and Other Antioxidation Systems

The antioxidant effect of zeaxanthin and vitamin E (tocopherol) in biological membranes are synergistic (i.e., multiplicative rather than additive; [56,57]; see also [58]). In addition, both zeaxanthin and vitamin E can be recycled by water-soluble antioxidants at the membrane-cytosol interface; the oxidized forms of zeaxanthin and vitamin E can both be re-reduced by water-soluble vitamin C [57] or water-soluble plant phenolics [59,60,61]. Oxidized vitamin C and other water-soluble antioxidant metabolites can, in turn, be re-reduced by antioxidant enzymes. Antioxidant enzymes require dietary mineral cofactors (e.g., copper, zinc, selenium) that is acquired from the soil by plants and passed on through the diet to humans. Zeaxanthin, lutein, and other antioxidants, furthermore, exhibit synergistic effects with polyunsaturated fatty acids, in particular omega-3 fatty acids. Lipid-peroxidation-based gene regulators in humans include hormones derived from omega-6 fatty acids that initiate inflammation, as well as hormones derived from omega-3 fatty acids that are potent terminators of inflammation [62]. A diet that provides a combination of synergistically acting antioxidant metabolites (e.g., xanthophylls, vitamin C, vitamin E), antioxidant minerals, and a high ratio of omega-3 to omega-6 fatty acids produces a strong anti-inflammatory effect.

While this review focuses on zeaxanthin and lutein, it should be noted that there is evidence for protective roles of other carotenoids as well. An example is the precursor lycopene (for a review see [63]) in the synthesis of β-carotene, zeaxanthin and lutein (Figure 1). Whereas lycopene does not accumulate in leafy greens, a block in the synthesis of the derived carotenoids causes strong lycopene accumulation in tomato. Tomato lycopene has been shown to lower the risk of multiple diseases, due to its antioxidant properties as an efficient quencher of singlet oxygen, and its effect in counteracting lipid peroxidation [63].

## 6. Dual Roles of Oxidants and Antioxidants

### 6.1. Enhancement of Eye and Brain Function beyond Disease

Oxidation of membrane lipids by reactive oxygen species and radicals was originally considered a purely deleterious process—but it is now clear that enzymatic lipid peroxidation gives rise to a host of derivatives that act as hormonal gene regulators [64], including the above-mentioned inflammation-terminating hormones [65]. Moreover, ROS provide input into a host of signal transduction networks, that control not only defense against invaders, but also multiple aspects of growth, development, and reproduction of living organisms, including humans [66,67] and plants [68]. While excess ROS can be damaging, moderate ROS levels are thus a vital source of information and defense in all life forms.

ROS serve as sensors of the state of the organism’s internal and external environment, communicate both threats and opportunities posed by its surroundings to the organism, and orchestrate responses at many levels within the organism (Figure 4; [68]). The balance between ROS and antioxidants changes over a continuum of intensity of external cues. For example, physical activity in humans generates ROS in the mitochondria of working muscles, which triggers the production of antioxidants (for a review, see [69]). Physical inactivity fails to provide the necessary ROS, and leads to low antioxidant levels (Figure 4). Excessively intense exercise can promote inflammation by exceeding the finite capacity to produce antioxidants (Figure 4; [70]). A similar scenario can be observed in plants with respect to light intensity, where low-light-grown plants possess minimal antioxidant capacity, while plants grown in high light have a full antioxidant complement (Figure 4; [71]); plants exposed to excessively high light can experience an imbalance between ROS and antioxidants and potential damage (Figure 4; [72]). For plants, conditions that can generate very high levels of ROS include not only excess light, but also heat or frost, either too little or too much water or nutrients, and presence of pests or pathogens. For humans, factors that generate very high levels of ROS include various forms of radiation, pollution, smoking, excess alcohol or drugs, as well as chronic psychological stress, excessive exercise, an excessively energy-dense, micronutrient-deficient diet, and infection with pathogens. Both plants and humans also produce ROS internally when under attack by pathogens, which mobilizes defenses that depend on signaling roles of ROS, as well as the actual use of ROS to destroy invaders, but can potentially lead to systemic inflammation, self-attack, and excessive programmed cell death in humans (such as photoreceptor cell death) or extensive programmed cell death in plants [73].

### 6.2. Positive and Negative Effects of Antioxidants

Since antioxidants oppose oxidants, and since oxidants can have both damaging and vital positive effects, it follows that antioxidants not only protect against potential damage, but also impact these vital processes. Balance is key, i.e., balance between oxidants and antioxidants that contribute to a balanced cellular redox state.

For the green leaf as one of the two focus systems of this review, the optimal redox balance, and thus the optimal level of antioxidant production, varies depending on environmental context (Figure 4; see also below). ROS have multiple and often opposing roles in essential plant functions [74,75,76], and the same appears to be true for antioxidants (see [77]). In some contexts, more antioxidation based on zeaxanthin-dependent photoprotection and/or vitamin E supported greater seed production (in environments with rapidly fluctuating light; [78]), superior plant defense (against an insect pest; [79]), greater water conservation [80], and greater heat tolerance [81]. However, in other contexts, it was a lowering of zeaxanthin-dependent photoprotection and/or vitamin E content that supported superior plant defense (to fungal and bacterial pathogens; [82,83]), greater plant biomass production [84], and greater heat tolerance by way of greater water loss and more evaporative cooling [77,85,86,87].

For the other focus system of this review, the human body, successful implementation of therapeutic interventions using antioxidants and anti-inflammatories may also require the definition of optimal concentrations for a range of specific scenarios. As for plants, the role of ROS and antioxidants in the human immune response is complex and highly context-specific. For example, the function of certain immune cells can be enhanced by low levels of ROS, and either suppressed or enhanced by higher ROS level [88], which presumably makes the impact of antioxidants similarly context-specific.

Commonly used, modestly dosed zeaxanthin and lutein supplements are presently considered safe (see [7,17] for specific concentrations). On the other hand, the toxicity of high-dose supplementation with xanthophylls has been demonstrated [89]. Undesirable effects of high-dose antioxidant supplements have also been demonstrated for other antioxidants. For example, high-dose vitamin C supplements prevented muscle repair and muscle building during athletic training ([90]; see also [69]). High-dose supplements of vitamin E or β-carotene resulted in greater, rather than reduced, incidence of chronic disease in several large human clinical trials (for a review, see [91]). Moreover, the optimal concentration of supplemental antioxidants (such as zeaxanthin and lutein) may vary with other factors that impact redox balance, such as age, genetic background, exposure to radiation, smoking or other toxins, the presence or absence of pathogens, exercise level, and psychological stress level. These connections could be explored to develop future personalized therapies, and would benefit from quick and at least partly nonintrusive technology to quantify individual status of xanthophylls (see [17]) and other antioxidants.

## Figures and Tables

**Figure 1 molecules-25-03607-f001:**
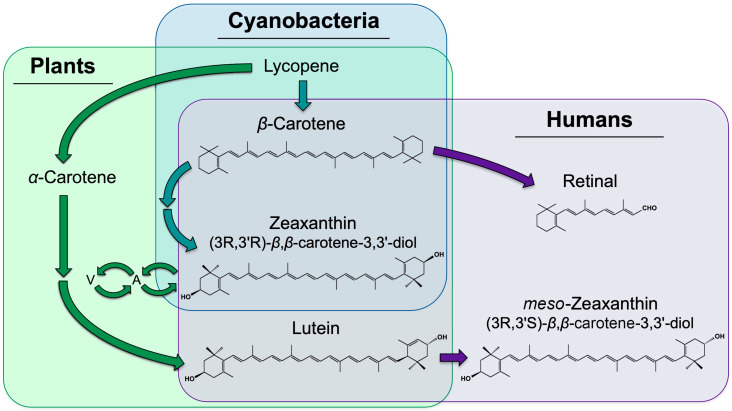
Biosynthetic pathways for the formation of zeaxanthin, lutein, and *meso*-zeaxanthin. Pathways existing in plants only, area shaded in green only; pathways existing in both plants and cyanobacteria, area shaded in both green and blue; pathways existing in humans, area shaded in purple only; need for dietary acquisition of β-carotene, zeaxanthin, and lutein by humans, overlapping purple and green/blue areas. A, antheraxanthin, a zeaxanthin mono-epoxide; V, violaxanthin, a zeaxanthin di-epoxide.

**Figure 2 molecules-25-03607-f002:**
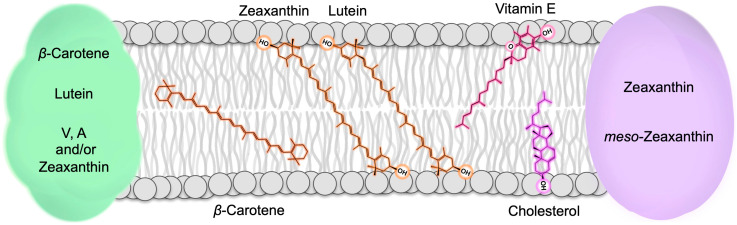
Schematic depiction of the localization of carotenoids, vitamin E (tocopherol), and cholesterol in biological membranes. Moreover, β-carotene, zeaxanthin, lutein, and tocopherols dissolve in the phospholipid biolayer of the membranes of both plants and humans/animals; cholesterol only occurs in animal membranes. In addition, binding of carotenoids to multiple components of light-collecting, chlorophyll-binding proteins in the photosynthetic membrane of plants (green oval) and of zeaxanthin and *meso*-zeaxanthin to a protein in the retinal membrane of humans (purple oval) is shown. A, antheraxanthin; V, violaxanthin.

**Figure 3 molecules-25-03607-f003:**
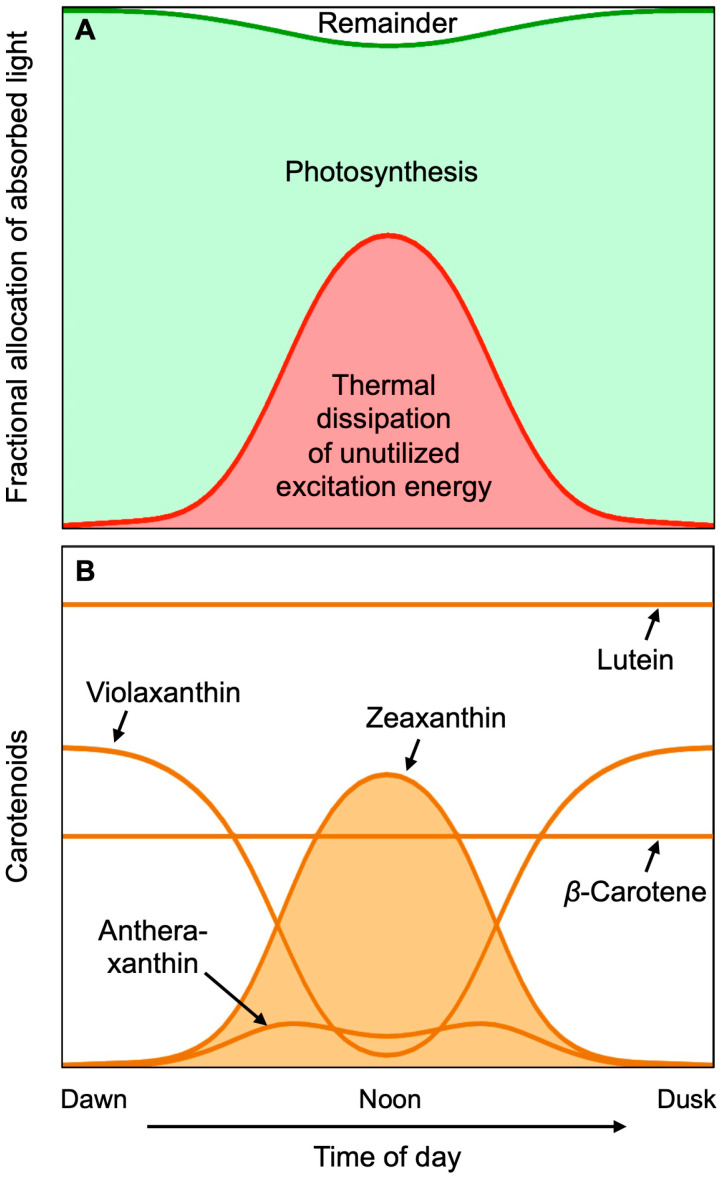
Schematic depiction of changes over the course of a clear day in (**A**) the fraction of light absorbed by a sun-exposed leaf that goes into photosynthesis (green), is dissipated harmlessly as heat (red), and is used by neither of the latter pathways (remainder; white). (**B**) shows corresponding changes in the level of the xanthophylls of the xanthophyll cycle, as well as the unchanged levels of lutein and β-carotene. Based on field data from evergreen shrubs reported in Adams et al. [18] and Demmig-Adams et al. [19]

**Figure 4 molecules-25-03607-f004:**
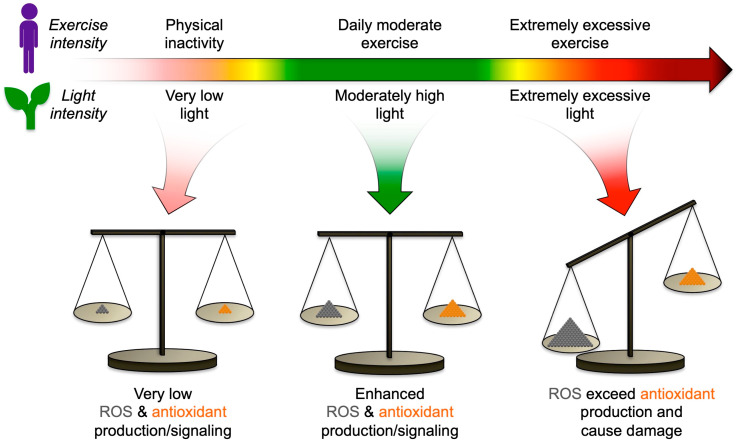
Schematic depiction of the proportion of reactive oxygen (ROS) and antioxidants produced across environmental gradients for the example of exercise intensity in humans and light intensity in plants.

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
