# Peer review of "Zeaxanthin and Lutein: Photoprotectors, Anti-Inflammatories, and Brain Food"

_molecules, 2020, doi:10.3390/molecules25163607_

Round 1
Reviewer 1 Report
This is a great review, very informative, very well designed and written, good drawings and most of the relevant references in the field.
Something useful for chemists as well as biologists active in this field. For someone slightly outside the field not easy to comprehend, but this is in the nature of the complex processes that are described herein.
The different roles and activities of carotenes and zeaxanthins / lutein in protective and activative functions and the balance between antioxidative positive and negative effects are convincingly described. I would like to read a little bit more about lycopene, the precursor to both plant and bacteria carotene/oid synthesis. Lycopene is described as the absolute winner in antioxidative power, e.g. the singlet oxygen quenching is two times stronger / faster than for ß-carotene. Is it worth mentioning this compound and its role in biosynthesis and antioxidative power in some sentences? I feel so.
Author Response
We have added the following statement and an additional citation:
"While this review focuses on zeaxanthin and lutein, it should be noted that there is evidence for protective roles of other carotenoids as well. An example is the precursor lycopene (for a review see [63]) in the synthesis of b-carotene, zeaxanthin and lutein (Figure 1). Whereas lycopene does not accumulate in leafy greens, a block in the synthesis of the derived carotenoids causes strong lycopene accumulation in tomato. Tomato lycopene has been shown to lower the risk of multiple diseases due to its antioxidant properties as an efficient quencher of singlet oxygen and its effect in counteracting lipid peroxidation [63]."
[63] Przybylska, S. Lycopene – a bioactive carotenoid offering multiple health benefits: a review. J. Food Sci. Tech. 2020, 55, 11–32. https://doi.org/10.1111/ijfs.14260
Reviewer 2 Report
Authors have collected appropriate references and wrote this review in detail.
- This review is focus on the zeaxanthin and its structural isomer lutein in plants and humans.
- This review exactly described the biosynthetic pathways of zeaxanthin, lutein, and meso-zeaxanthin and the effect of light duality on their biosynthesis.
- This review also described the benefit effect of zeaxanthin, lutein, and meso-zeaxanthin on plant physiology and their interaction and transport with bilayer membranes.
- This review also described the transport, storage and conversion of zeaxanthin, lutein, and meso-zeaxanthin in human, and the benefit effects on eyes and brain function.
- This review also described the synergic effects, safety and toxicology of zeaxanthin, lutein, and meso-zeaxanthin as a diet supplement.
Author Response
We have read the reviewer's comments and are grateful for the appreciation of our approach.
Reviewer 3 Report
The paper by B. Demmig-Adams and coauthors is a very clearly written review. I can recommend this paper for publication and I have only one suggestion.
In page 3, lines 73-75, the sentence “While zeaxanthin orients itself perpendicularly to the plane of the membrane, lutein can apparently alternate between perpendicular and parallel positions [8] ” should be clarified. In my opinion there is no difference in the orientation of lutein and zeaxanthin in the lipid bilayer. Both adopt a similar transmembrane orientation. A recent publication from Gruszecki’s group corrected their previous results [8], which indicated that lutein has perpendicular and parallel orientation. I suggest to the authors to refer to
Grudzinski, W.; Nierzwicki, L.; Welc, R.; Reszczynska, E.; Luchowski, R.; Czub, J.; Gruszecki, W.I. Localization and Orientation of Xanthophylls in a Lipid Bilayer. Scientific Reports 2017, 7, 1–10, doi:10.1038/s41598-017-10183-7.
Author Response
We are grateful for the reviewer's very helpful suggestion, and have added the following statement as well as the additional reference pointed out by the reviewer.
"Zeaxanthin and lutein orient themselves across the membrane as shown in Figure 2 [8]. Whereas all zeaxanthin and lutein was located within membranes in a single-bilayer system [8], some lutein, but no zeaxanthin, was also found in the inter-membrane region of multiple-bilayer systems [9]. It has been suggested that the intra-membrane position allows xanthophylls to serve as “molecular rivets” that enhance membrane structural integrity [10]"
[8] Grudzinski, W.; Nierzwicki, L.; Welc, R.; Reszczynska, E.; Luchowski, R.; Czub, J.; Gruszecki, W.I. Localization and orientation of xanthophylls in a lipid bilayer. Rep. 2017, 7, 9619. https://doi.org/10.1038/s41598-017-10183-7